# Architecture of the chromatin remodeler RSC and insights into its nucleosome engagement

Avinash B Patel[1,2†]*, Camille M Moore[3†], Basil J Greber[2,4†], Jie Luo[5], Stefan A Zukin[6], Jeff Ranish[5], Eva Nogales[2,3,4,7]*

[1]Biophysics Graduate Group, University of California, Berkeley, Berkeley, United States; [2]Molecular Biophysics and Integrative Bio-Imaging Division, Lawrence Berkeley National Laboratory, Berkeley, United States; [3]Molecular and Cell Biology Department, University of California, Berkeley, Berkeley, United States; [4]California Institute for Quantitative Biology (QB3), University of California, Berkeley, Berkeley, United States; [5]The Institute for Systems Biology, Seattle, United States; [6]Chemistry Department, University of California, Berkeley, Berkeley, United States; [7]Howard Hughes Medical Institute, University of California, Berkeley, Berkeley, United States

**Abstract** Eukaryotic DNA is packaged into nucleosome arrays, which are repositioned by chromatin remodeling complexes to control DNA accessibility. The *Saccharomyces cerevisiae* RSC (Remodeling the Structure of Chromatin) complex, a member of the SWI/SNF chromatin remodeler family, plays critical roles in genome maintenance, transcription, and DNA repair. Here, we report cryo-electron microscopy (cryo-EM) and crosslinking mass spectrometry (CLMS) studies of yeast RSC complex and show that RSC is composed of a rigid tripartite core and two flexible lobes. The core structure is scaffolded by an asymmetric Rsc8 dimer and built with the evolutionarily conserved subunits Sfh1, Rsc6, Rsc9 and Sth1. The flexible ATPase lobe, composed of helicase subunit Sth1, Arp7, Arp9 and Rtt102, is anchored to this core by the N-terminus of Sth1. Our cryo-EM analysis of RSC bound to a nucleosome core particle shows that in addition to the expected nucleosome-Sth1 interactions, RSC engages histones and nucleosomal DNA through one arm of the core structure, composed of the Rsc8 SWIRM domains, Sfh1 and Npl6. Our findings provide structural insights into the conserved assembly process for all members of the SWI/SNF family of remodelers, and illustrate how RSC selects, engages, and remodels nucleosomes.

*For correspondence:
patelab@berkeley.edu (ABP);
enogales@lbl.gov (EN)

†These authors contributed
equally to this work

Competing interests: The
authors declare that no
competing interests exist.

Reviewing editor: Cynthia
Wolberger, Johns Hopkins
University School of Medicine,
United States

## Introduction

Eukaryotes have four major families of chromatin remodelers: SWI/SNF, ISWI, CHD, and INO80 (*Clapier and Cairns, 2009*). Each of these remodelers plays distinct roles based on how they select and affect target nucleosomes. Together, these remodelers give rise to the distinct chromatin landscapes observed in eukaryotic cells and determine how genetic information is organized, replicated, transcribed, and repaired (*Yen et al., 2012*). In *S. cerevisiae* there are two members of the SWI/SNF family of chromatin remodelers: RSC and SWI/SNF (*Côté et al., 1994*; *Cairns et al., 1996*). SWI/SNF chromatin remodelers reposition nucleosomes by translocating DNA around the histone octamer, and in vitro assays have shown that they move nucleosomes to the ends of linear DNA fragments before evicting the histones from the DNA (*Clapier et al., 2016*). RSC is essential for yeast viability and is ten times more abundant than SWI/SNF (*Cairns et al., 1996*). In the context of transcription, RSC is responsible for maintaining nucleosome free regions (NFR), while SWI/SNF plays a role in remodeling nucleosomes during transcription initiation (*Nagai et al., 2017*; *Krietenstein et al., 2016*; *Klein-Brill et al., 2019*). Additionally, RSC is also involved in many transcription-independent

processes, such as mitotic division, double stranded break repair, and telomere maintenance (*Krietenstein et al., 2016*; *Erkina et al., 2010*; *Kuryan et al., 2012*; *Campsteijn et al., 2007*; *Shim et al., 2007*; *Ungar et al., 2009*; *Ng et al., 2002*).

RSC has a molecular weight of ~1.1 MDa and is composed of 17 proteins, with two copies of Rsc8 and one copy of either Rsc1 or Rsc2 (*Cairns et al., 1996*; *Cairns et al., 1999*). Seven of the proteins (Arp7, Arp9, Rsc6, Rsc8, Sfh1, Sth1 and Rsc9) are conserved in all eukaryotic SWI/SNF chromatin remodeling complexes, with Sth1 containing the ATPase domain that is responsible for the remodeling activity (*Cairns et al., 1996*; *Saha et al., 2002*). Two additional subunits are conserved in yeast (Npl6 and Rtt102), and the rest are complex specific (Htl1, Ldb7, Rsc1/2, Rsc3, Rsc30, Rsc4 and Rsc58). Three subunits, Arp7, Arp9 and Rtt102, are shared between RSC and SWI/SNF, and the two Arps have been found to be important for efficient remodeling activity of RSC (*Clapier et al., 2016*). The SWI/SNF paralog of Rsc8, Swi3, has been shown to be important for complex assembly and to likely play a scaffolding role (*Yang et al., 2007*), while the paralog of Sfh1, Snf5, has been found to be important for remodeling activity (*Sen et al., 2017*). RSC contains six bromodomains (BrDs), two in RSC1/2 and Rsc4, and one in Rsc58 and Sth1. Additionally, Rsc1/2 also contain a BAH domain. The BrD and BAH domains of RSC have all been shown to interact with histones H3 (*Chambers et al., 2013*), with the BrDs favoring acetylated H3 tails (*Zhang et al., 2010*), and in vitro binding assays have shown that the affinity of RSC for acetylated H3 nucleosomes is indeed greater than for unmodified nucleosomes (*Chatterjee et al., 2011*). In vivo deletions or mutations in these domains have been shown to decrease cell survival (*Cairns et al., 1999*; *Kasten et al., 2004*). The fungal-specific subunits Npl6, Htl1, Ldb7, Rsc3 and Rsc30, have been proposed to form a structural module (*Wilson et al., 2006*), with Npl6 and Htl1 interacting with Rsc8, and Ldb7 with Sth1 and Arp9. Rsc3 and Rsc30, which dimerize with each other, share a very similar architecture (*Angus-Hill et al., 2001*), with an N-terminal Zinc cluster that has been shown to bind to the DNA sequence CGCG that preferentially occurs at gene promoters (*Badis et al., 2008*). Deletion of Rsc3 has been shown to weaken the ability of RSC to maintain NFR in vivo (*Badis et al., 2008*).

High-resolution structural studies of the SWI/SNF family of remodelers had until recently been limited to fragments, such as the Arp module (Arp7, Arp9 and some combination of Rtt102 and HSA helix of Snf2 [paralog to Sth1]) (*Schubert et al., 2013*), the SwiB domain of the human SMARCD (homolog of Rsc6), the WH domain of SMARCB (homolog of Sfh1) (*Allen et al., 2015*), the SWIRM domain of Swi3 (paralog of Rsc8) (*Da et al., 2006*), the RPT domain of SMARCB (homolog of Sfh1) (*Sammak et al., 2018*), the SWIRM-RPT complex of SMARCC-SMARCB (homologs of Rsc8 and Sfh1) (*Yan et al., 2017*), the SANT domain of SMARCC (homolog of Rsc8), and the ATPase of Snf2 (homolog of Sth1) on its own (*Dürr et al., 2005*; *Xia et al., 2016*) and in complex with a nucleosome with different ATP analogs (*Liu et al., 2017*; *Li et al., 2019*). Early structural studies of full SWI/SNF remodeling complexes by negative stain electron microscopy (EM) were limited by low resolution and/or reconstruction artifacts (*Chaban et al., 2008*; *Dechassa et al., 2008*; *Asturias et al., 2002*; *Leschziner et al., 2007*; *Leschziner et al., 2005*; *Skiniotis et al., 2007*; *Smith et al., 2003*), with only one study able to map the location of some subunits within the complex (for the yeast SWI/SNF complex using subunit deletion) (*Zhang et al., 2018*).

Here, we have used cryo-EM to determine the structure RSC. The core of the complex was resolved to 3 Å, which allowed us the de novo building of this entire region. The complex is scaffolded around a central Rsc8 dimer from which other evolutionarily conserved subunits assemble, leading to a model for the biogenesis of the complex that agrees with that previously proposed for human SWI-SNF complexes based on biochemical data (*Mashtalir et al., 2018*). We were also able to determine the structure of RSC bound to a nucleosome at 19 Å resolution, which, together with our structure of the core of RSC, and previous ones for the Arp module and nucleosome, allowed us to reveal how RSC engages nucleosomes.

## Results

### Structure of RSC and RSC-NCP

We have used cryo-EM to determine the structure of the chromatin remodeler RSC from *S. cerevisiae* purified using the TAP-tag (on Sth1) method (*Figure 1—figure supplement 1*). We found that RSC is composed of five main lobes, three that form a relatively rigid core (head, body and arm) and

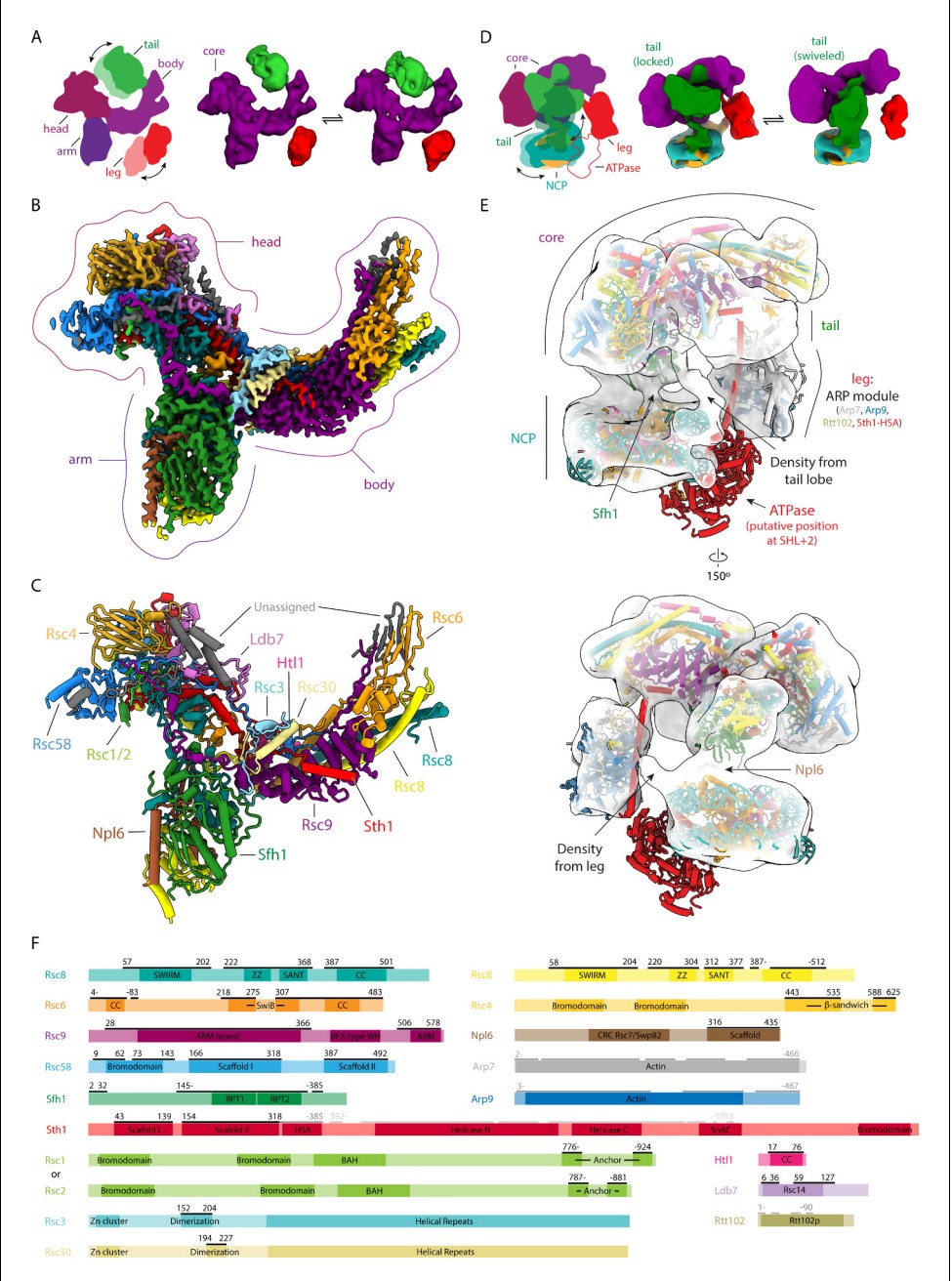

**Figure 1.** Structures of RSC and RSC-NCP complex. (**A**) On the left, cartoon representation of RSC showing its five main lobes. The head, body and arm lobes form the core of the complex, while the tail and leg lobes are flexible. On the right are two cryo-EM reconstructions of RSC with the tail (green) and leg (red) lobes in two different conformations with respect to the core (purple). (**B**) Cryo-EM reconstruction of the RSC core with individually subunits colored. (**C**) Model of the RSC core with individual subunits colored and labelled. (**D**) On the left, cartoon representation of the RSC-NCP complex showing the five main lobes of RSC colored as in A, and with DNA in teal and histones in orange. The tail and leg lobes are flexible. On the right are two cryo-EM reconstructions of RSC-NCP showing the tail lobe in two different conformations with respect to the core. (**E**) Cryo-EM reconstruction of RSC-NCP in transparent with the core of RSC, the NCP (PDB: 6IY2) (*Li et al., 2019*) and the ARP module (PDB: 4I6M) (*Schubert et al., 2013*) structures docked in. The ATPase domain (not visible in our density) is modeled according to the structure of nucleosome-bound Snf2 (PDB: 6IY2) (*Li et al., 2019*). The points of contact between RSC and the NCP are labeled. (**F**) Domain maps for RSC subunits. Regions modeled within the RSC core are marked by black lines above the schematic of each protein. The ARP module that was docked into the RSC-NCP map is marked by the dark grey lines and the ATPase domain as shown in the model is marked by light grey lines.

*Figure 1 continued on next page*

*Figure 1 continued*

The online version of this article includes the following figure supplement(s) for figure 1:

**Figure supplement 1.** Biochemical Characterization of RSC.
**Figure supplement 2.** Negative stain analysis of SWI/SNF and RSC.
**Figure supplement 3.** Cryo-EM data collection and processing for apo RSC.
**Figure supplement 4.** Domain maps of RSC subunits.
**Figure supplement 5.** Modeling of RSC core.
**Figure supplement 6.** Modeling of Rsc1/2.
**Figure supplement 7.** RSC core structure-model validation.
**Figure supplement 8.** Chemical Crosslinking Mass Spectrometry Analysis of RSC.
**Figure supplement 9.** Cryo-EM data collection and processing for the RSC-NCP complex.

two that are flexibly attached (leg and tail) (*Figure 1A*; *Figure 1—figure supplement 2*). Our negative stain analysis of the yeast SWI/SNF complex shows that, like RSC, it has a rigid core sharing some similar features, and a flexible leg that occupies similar overall positions (*Figure 1—figure supplement 2*). However, SWI/SNF lacks a tail lobe, indicating that while RSC and SWI/SNF share a conserved core architecture, RSC features additional regulatory domains (*Figure 1—figure supplement 2*). Our assignment of most of the tail region to the RSC-specific subunits Rsc3 and Rsc30 (see below) is in further agreement with this observation.

We determined the structure of the core to ~3.0 Å and mapped 14 proteins within this region: Rsc1/2, Rsc3, Rsc4, Rsc6, Rsc8 (two copies), Rsc9, Rsc30, Rsc58, Ldb7, Npl6, Htl1, Sfh1, and Sth1 (*Figure 1B,C,F*; *Figure 1—figure supplements 3*, *4*, *5*, *6* and *7*). It was not possible to distinguish, based on our cryo-EM data, whether the structure contained Rsc1 or Rsc2. A region in our map could be mapped either to Rsc1 (776–809 and 892–924) or Rsc2 (787–820 and 849–881). These regions are highly similar between the two proteins (43% identical plus 22% similar) and the density we observe likely correspond to an average of these two proteins (*Figure 1—figure supplement 6*).

We confirm our model by mass spectrometry analysis of RSC chemically crosslinked using bis(sulfosuccinimidyl)suberate (BS3) (*Figure 1—figure supplement 8*). We identified 780 unique intermolecular links between different subunits and 617 unique intra-molecular links within the same subunits. About 90% (151/168) of mappable crosslinks in our model of the RSC core structure are within 38 Å distance.

In order to shed light on the interaction of RSC with its substrate, we also obtained a 19 Å resolution map of RSC bound to a nucleosome core particle (NCP) modified with H3K4me3 and H3K(9/14/18)ac (*Figure 1D,E,F*; *Figure 1—figure supplement 9*). We used an acetylated nucleosome due to the higher affinity of RSC for nucleosomes containing acetylated H3 (*Chatterjee et al., 2011*). We were able to unambiguously fit our structure of the RSC core within this map, along with previously published models of the Arp module (Arp7, Arp9, Rtt102 and Sth1-HSA) and the NCP (*Figure 1E*) (*Schubert et al., 2013*; *Li et al., 2019*). We do not observe any density for the ATPase domain of Sth1; however, based on the structure of the Sth1 homolog Snf2 bound to the NCP, we expect the catalytic subunit to bind super-helical location-2 (SHL2) (*Li et al., 2019*).

Our data show two distinct modes of NCP binding by RSC, one where the whole tail is swiveled towards the nucleosome (termed swiveled), and one where only a small region emanating from the tail reaches towards the NCP (termed locked) (*Figure 1D*). Altogether, the core of RSC appears to bind the nucleosome in a well-defined orientation that in our docking-based model places the catalytic domain of Sth1 in a position that would allow it to interact with the SHL2. The fact that we observe nucleosome-bound RSC, even without a stable binding of the catalytic domain of Sth1 to the NCP, indicates that the contacts through the RSC core are sufficient for nucleosome engagement. It also makes our structure a likely intermediate in a pathway to full engagement of a nucleosome by RSC. We propose that the contacts made by the RSC core with the NCP contribute to the processivity of the nucleosome remodeling function of the catalytic domain of Sth1 (see below).

## Structural organization of RSC

The RSC core is critically defined by subunits Rsc8, Rsc6, Rsc9, Rsc58, Sth1, and Sfh1. Five of these proteins (Rsc8, Rsc6, Rsc9, Sth1, and Sfh1) are evolutionarily conserved throughout the eukaryotic SWI/SNF family, and comprise 72% of the mass of the core density (*Figure 2A*; *Figure 2—source*

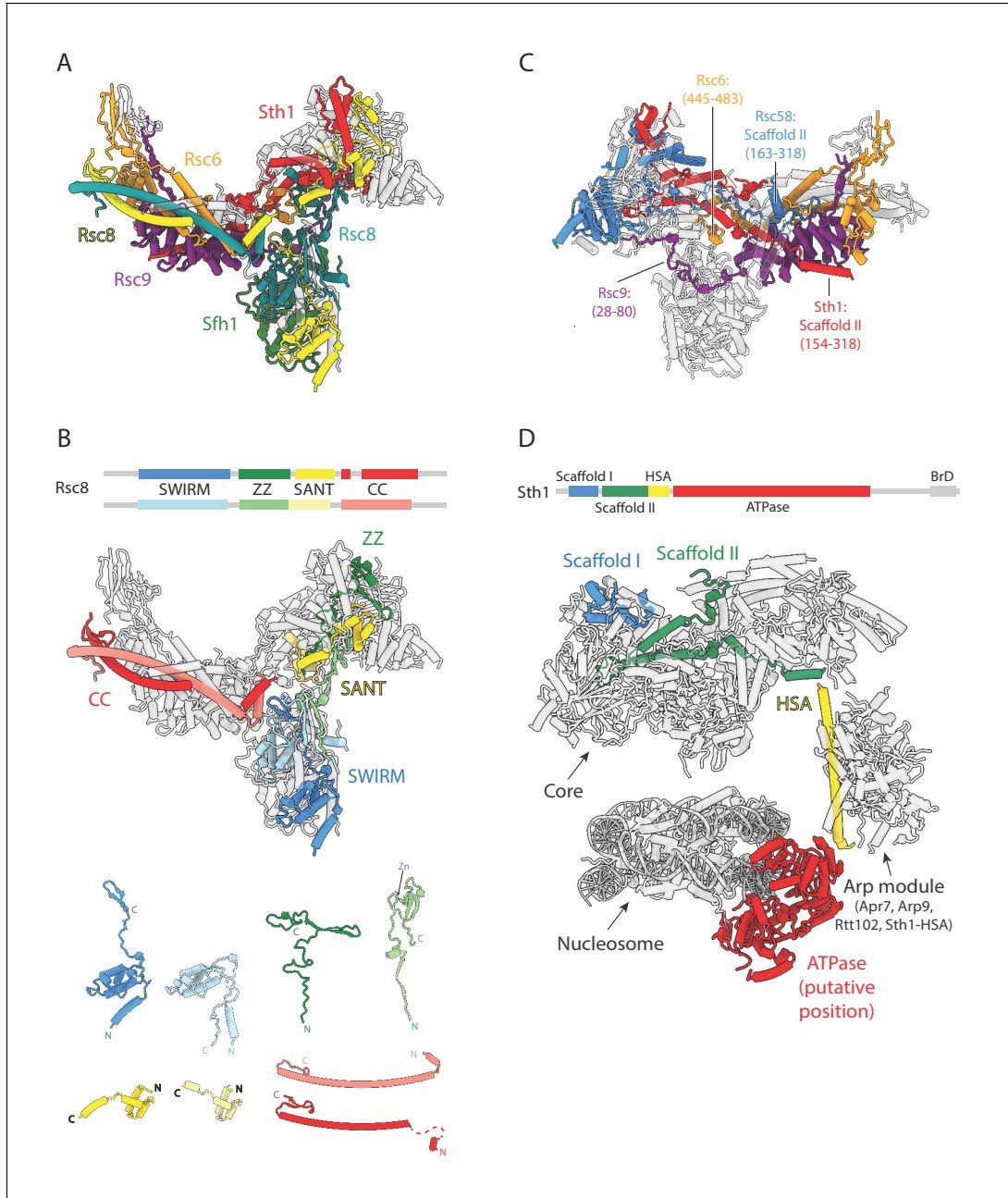

**Figure 2.** Conserved structural scaffold in RSC. (A) Structure of the RSC core, highlighting in color the evolutionarily conserved subunits. Non-conserved subunits are shown in transparent grey. Subunit labels are shown with colors corresponding to those in the structure. (B) Top, domain map of Rsc8, with the four different domains colored blue to red from N to C terminus. Middle, structure of the RSC core with only the Rsc8 dimer colored. Bottom, individual domains of the two copies of Rsc8, aligned and tiled. (C) RSC subunits that span multiple lobes (except for Rsc8) are colored, with the regions spanning them labeled. (D) Model of the RSC-NCP complex with domains of Sth1 highlighted blue to red from N to C terminus. Scaffolding domains I (blue) and II (green) interact with the core region of RSC, while the HSA helix (yellow) interacts with the Arp module to form the leg lobe. The ATPase (red; not visible in our density) is modeled according to the structure of nucleosome-bound Snf2 (ref. *Li et al., 2019*). The online version of this article includes the following source data for figure 2:

**Source data 1.** SWI/SNF family of chromatin remodelers.

*data 1*) (*Wang et al., 1996*; *Kadoch and Crabtree, 2015*). An asymmetric Rsc8 dimer defines the backbone for the complex, scaffolding the three core lobes and contacting all core proteins except Rsc3 and Rsc30 (*Figure 2A,B*). Rsc8 has four conserved domains, which are, from N to C terminus:

SWIRM, ZZ, SANT, and coiled coil (CC) (*Figure 2B*). The SWIRM domains are in the arm, the ZZ and SANT domains in the head and the CC domains in the body. Previous work has shown that Rsc8 homologs are critical for the integrity of their respective SWI/SNF chromatin remodelers, supporting the idea that Rsc8 and its homologs are the structural backbone for all SWI/SNF family of remodelers (*Mashtalir et al., 2018*).

Apart from Rsc8, the only other subunits to span multiple lobes in our model of the core are Rsc6, Rsc9, Rsc58 and Sth1 (*Figure 2C*). Most of the structured regions of Rsc6 are in the body, except for a small region at the very C-terminus of the protein (residues 445–483), which occupies the space in between the three lobes. Rsc9 also primarily located in the body, but for a small region at the very N-terminus terminus (residues 28–80) interacts with the surface of the arm and head lobes. The N-terminal BrD and C-terminal Scaffold II domain of Rsc58 are in the head lobe, while the central Scaffold I domain is in the body lobe. Sth1 spans three lobes, the head, body and leg. The Scaffold I domain is in the head while the Scaffold II domain goes between the body to the head and back to the body. From the RSC-NCP model, we observe that Sth1 then continues from the Scaffold II domain to the HSA helix that, along with Arp7, Arp9 and Rtt102, form the leg lobe (*Figure 2D*). From the HSA helix, Sth1 continues into the ATPase domain that would putatively bind the nucleosome at position SHL2 (*Figure 2D*). This model of RSC would have the N- and C- termini of adjacent Sth1 domains at the right distance to connect through the short linkers between them (*Figure 2D*).

## Architecture of the body lobe

The body lobe of RSC contains a helical bundle backbone, an α-solenoid belly and a β-sandwich hook (*Figure 3A*). The helical bundle is made up of the two CC domains of Rsc8, along with the CC domains of Rsc6 and Htl1. This helical bundle interacts with the α-solenoid belly made up of the armadillo repeats of Rsc9 and the SwiB domain of Rsc6. Finally, the hook region off the belly contains β-strands from Rsc9 and Rsc6 that come to together to form a b-sandwich. Given the position of the Rsc9 sequences model in our structure, it is likely that the RFX-type WH of Rsc9 is part of the tail. In addition to these major structural regions there are some smaller elements that contribute to the body lobe. These include a segment of Rsc58 that binds between the belly and the backbone, a small segment of Sth1 that binds to the surface of the helical bundle, and a small segment of Sth1 that binds the surface of the armadillo repeat (*Figure 3A*). The last two components of the body are the Rsc3 and Rsc30 dimer that binds the surface of the Rsc9 armadillo repeat on top of Sth1 (*Figure 3A*). The Rsc3 and Rsc30 dimer appears to be absent in 75% of the RSC particles that were used in the high-resolution analysis (*Figure 3B*), in agreement with previous findings showing that some RSC complexes lack Rsc3 and Rsc30 (*Angus-Hill et al., 2001*). During our image analysis, we found that the presence or absence of the Rsc3 and Rsc30 dimer correlates well with the presence or absence of most of the tail lobe, making the Rsc3 and Rsc30 subunits the most likely constituents of this very flexible lobe (*Figure 3C*). The fact that there are no equivalent subunits in the yeast SWI-SNF complex, for which the tail domain is missing, further supports this assignment (*Figure 1—figure supplement 2*).

## The arm lobe engages the NCP

Four proteins contribute to most of the arm lobe, which is made of the two SWIRM domains of Rsc8 held together by the Scaffold domain of Npl6 and the RPT1 and RPT2 domains of Sfh1 (*Figure 4A*). Within our RSC-NCP structure, the arm lobe makes major interactions with the nucleosome, contacting both the acidic patch and nucleosomal DNA (*Figure 4B*). The region of the arm that contacts the acidic patch appears to emanate from the C-terminal end of the RPT2 domain of Sfh1 (*Figure 4B*). This region of Sfh1 is highly conserved and contains nine lysine and arginine residues that are predicted to form a helix (*Figure 4C*). The involvement of this region in nucleosome binding is supported by the fact that deletion of Snf5 (SWI/SNF homolog of Sfh1) in yeast results in less efficient chromatin remodeling activity of the SWI/SNF complex (*Sen et al., 2017*).

The region of the arm lobe contacting the nucleosomal DNA appears to emanate from either the C-terminus of the SWIRM domain of Rsc8, which is predicted to be disordered, or the N-terminus of the Scaffold domain of Npl6 (*Figure 4B*), which includes a CRC domain which shares homology to the WH domain of SMARCB (*Allen et al., 2015*; *Söding et al., 2005*). The WH domain of SMARCB

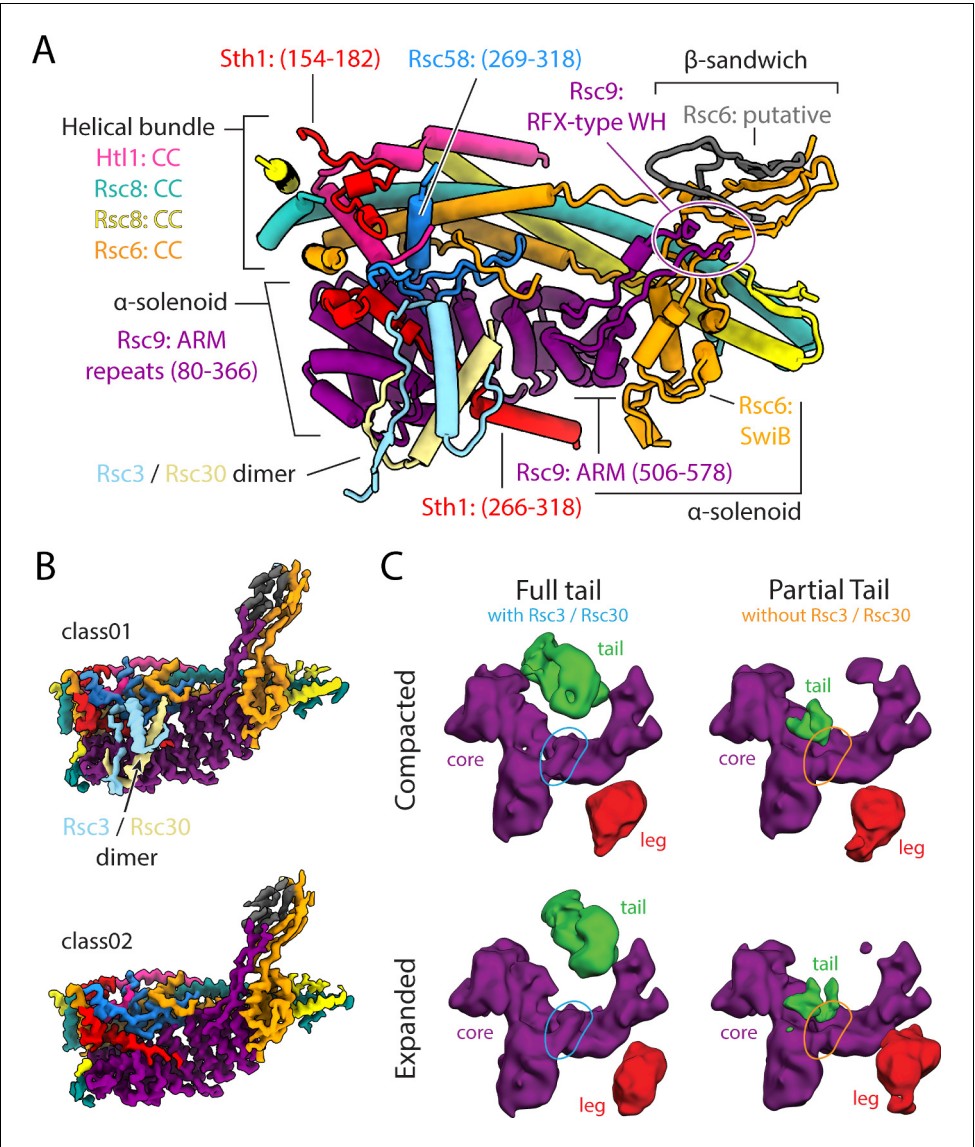

**Figure 3.** Architecture of the body lobe and occupancy of the tail lobe. (**A**) Structure of the RSC body lobe, with protein domains and segments labeled. (**B**) Two maps generated from 3D classification of the body region. Shown is the presence or absence of RSC3/30 density with the body lobe. Maps were generated using partial signal subtraction and 3D classification. (**C**) Four classes generated from 3D classification using the final high-resolution set of particles, without mask and with a reference that was low pass filtered to 20 Å at every iteration. These classes reveal the flexibility of the tail and leg lobes of RSC and correlate the presence of the tail lobe with the Rsc3/Rsc30 occupancy in the core.

has been shown to bind DNA and would be small enough to be accommodated within the extra density observed in our RSC-NCP structure, making the CRC domain of Npl6 the likely candidate for binding the nucleosomal DNA (*Figure 4B*) (*Allen et al., 2015*).

## The head lobe contains chromatin interacting subunits

Six RSC subunits contribute majorly to the head lobe of RSC, which is made of the ZZ and SANT domains of the two copies of Rsc8, the BrD and Scaffold II domain of Rsc58, the Scaffold I and part of Scaffold II domain of Sth1, the β-barrel domain of Rsc4 and the anchor domain of Rsc1/2 (*Figure 5A*). Of these proteins, only Rsc8 and Sth1 are evolutionarily conserved, suggesting that this region probably contributes to specific functions of the RSC complex. Rsc58, Rsc4 and Rsc1/2 are

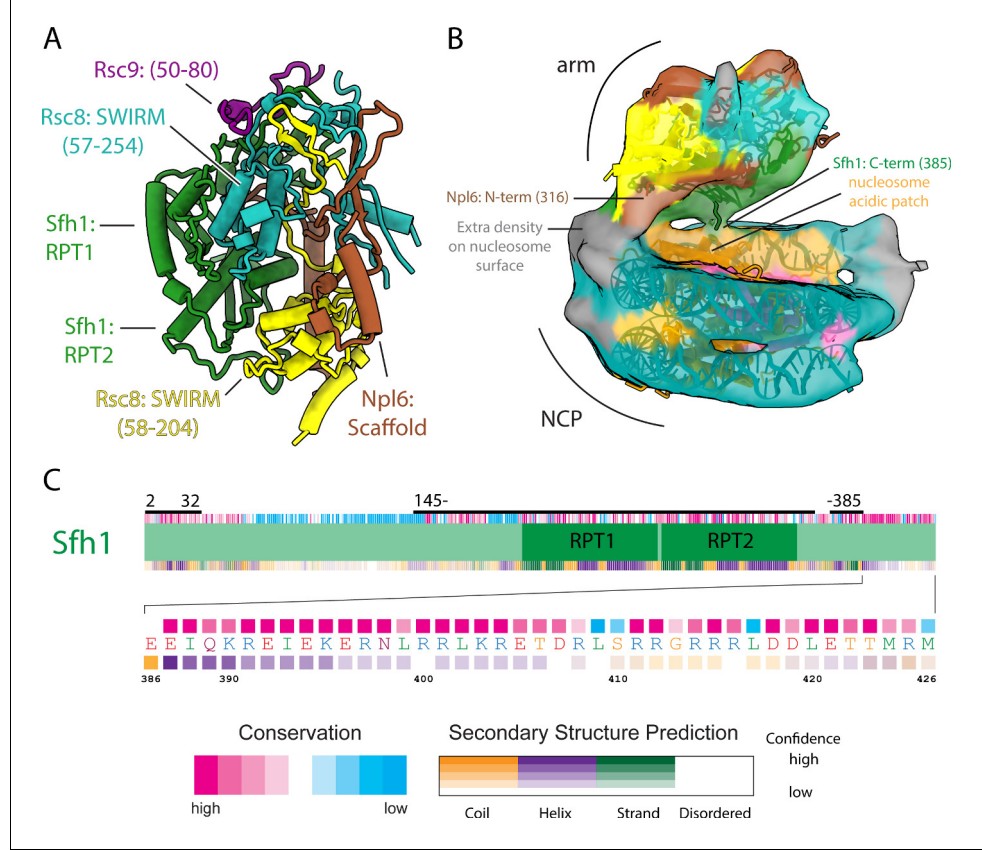

**Figure 4.** Architecture of the arm lobe and is interaction with the NCP. (**A**) The structure of the RSC arm lobe with protein domains and segments labeled. (**B**) The density of the NCP and arm region from the RSC-NCP reconstruction showing the connections that occur between the two. (**C**) Domain organization, sequence conservation and secondary structure prediction for Sfh1. Below is a zoomed in view of the C-terminus showing the sequence at a residue level. The domain map coloring is the same as in (*Figure 1—figure supplement 4*).

notable for containing histone reader domains: BrD and BAH domains. Only one other protein within RSC, Sth1, has a histone reader domain (a BrD). These domains have been shown to interact with acetylated lysines on histone tails, particularly H3 (*Chambers et al., 2013*; *Chatterjee et al., 2011*; *Duan and Smerdon, 2014*). We could only model the BrD of Rsc58, while the remaining BrDs and the BAH domain extend from flexible linkers and are not visible in our structure (*Figure 5A,B*). Based on the position of the head relative to the arm lobe, the BrD-containing subunits in our RSC-nucleosome model are positioned so that these domains could reach the H3 tails of the engaged nucleosome (*Figure 5B*). Alternatively, it is possible that the BrDs of one RSC complex could interact with different, adjacent nucleosomes. Microarray data have shown that different bromodomains within RSC can recognize different acetylation sites, which could allow the complex to be targeted to a wide variety of genomic loci (*Zhang et al., 2010*; *Filippakopoulos et al., 2012*). The yeast SWI/SNF and its human homolog BAF have just one BrD – on the ATPase subunit – while human PBAF has eight, indicating that these domains may contribute to functional specificity of different classes within the SWI/SNF family.

## Discussion

### Mechanistic model for nucleosome engagement by RSC

Based on our structure and existing biochemical data, we propose a 4-step mechanism of RSC-nucleosome engagement (*Figure 6*). Initial recruitment of RSC to a nucleosome likely occurs through interaction with H3 acetylation marks, given that RSC has an increased affinity for these nucleosomes

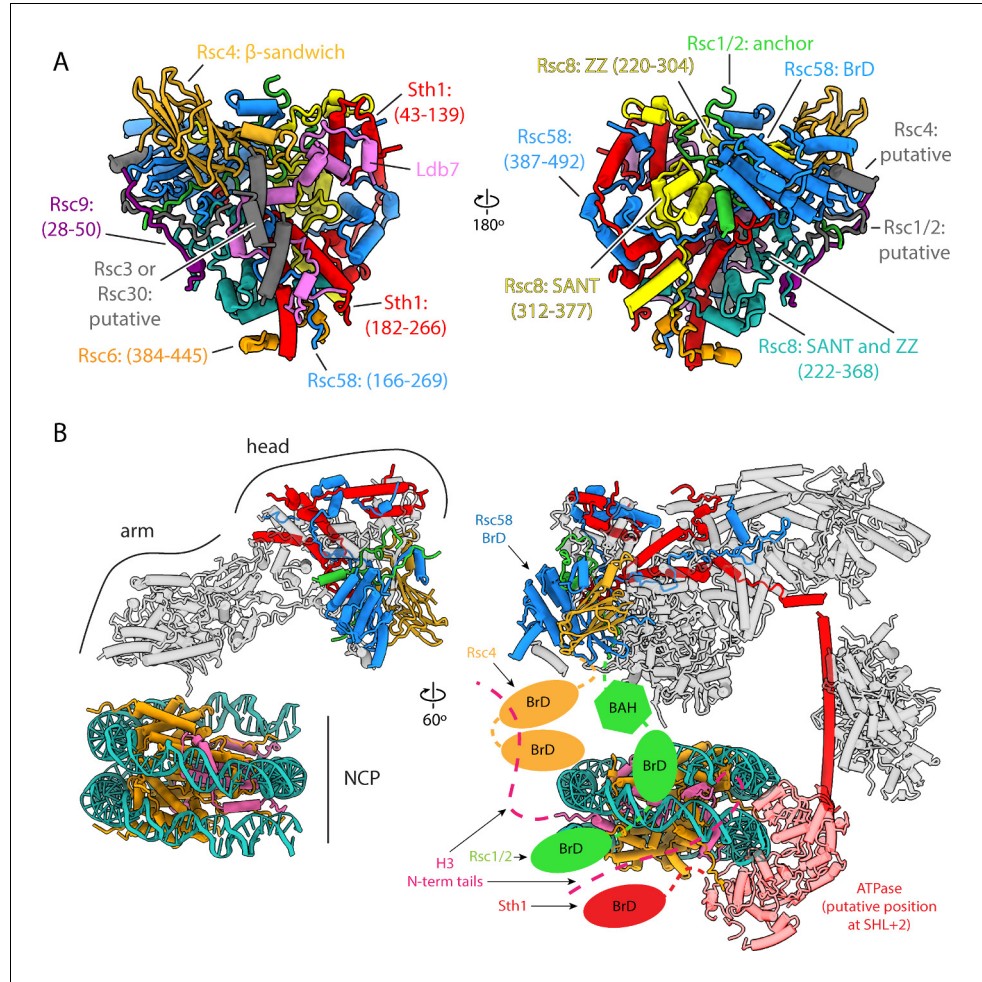

**Figure 5.** Architecture of the head lobe, the chromatin reader hub. (**A**) Structure of the RSC head lobe with protein domains and segments labeled. (**B**) On the left the RSC-NCP model showing the head lobe situated over the top of the nucleosome. On the right in the full RSC-NCP model with cartoon connections for the chromatin interacting domains (BrDs and BAH) shown. Only the RSC subunits that contain chromatin interacting domains are colored. The nucleosome is colored teal for DNA, orange for histones H2A, H2B and H4, and pink for H3.

(*Kasten et al., 2004*) and that most of them as flexible, facilitating an initial encounter. RSC could then engage the nucleosome through its arm region and place the SHL2 site of the nucleosome in position to bind the flexibly tethered catalytic domain of Sth1. The ATPase domain would then be able to translocate the DNA around the histone octamer, while the core of RSC holds onto the histone core through its direct interactions with histones. This process is very likely to apply to all SWI/SNF chromatin remodelers, as they share the same architecture and contain the conserved subunits/domains involved in the steps just proposed. Specifically, all complexes have bromodomains to bind acetylated histones (minimally in the C-terminus of their catalytic subunit), a Sfh1 homolog to bind the acidic patch of the nucleosome, and a catalytic DNA translocase domain to remodel nucleosomes (*Figure 2—source data 1*) (*Clapier and Cairns, 2009*).

## Model of RSC core assembly

Our structure of the RSC core allows us to visually represent the architecture of assembly intermediates of human SWI/SNF complexes that were previously identified based by biochemical and mass-spectrometry experiments (*Figure 7*) (*Mashtalir et al., 2018*) and to rationalize the assembly pathway based on the molecular contacts observed in our structure. According to this analysis, dimerization of the Rsc8 CC domains is most likely the first step in RSC complex assembly, followed by the binding of Rsc6 to this CC dimer. The RPT1 and RPT2 motifs within Sfh1 would then bring the two

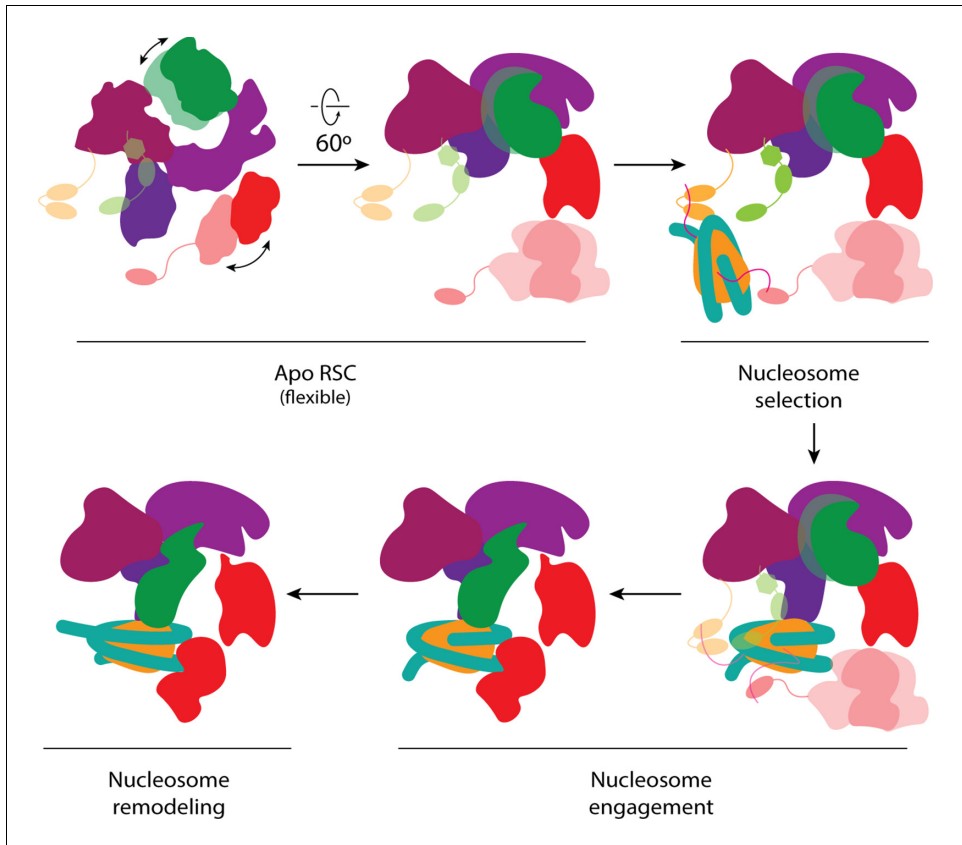

**Figure 6.** Proposed model of nucleosome engagement. Mechanistic model of RSC engaging a nucleosome. The apo RSC is shown with its moving tail and leg lobes, and its flexibly attached histone-tail binding domains. The BrDs bind and select target nucleosomes with acetylated tails. The selected nucleosome engages RSC, first through the arm lobe of the core, which then positions the ATPase domain at the end of the leg to be able to bind SHL2. During nucleosome remodeling, the ATPase translocates the DNA while the RSC core holds onto the histone core.

SWIRM domains of Rsc8 together. This initial order of assembly is based on the assembly intermediates identified for the mammalian SWI/SNF complexes BAF and PBAF (*Mashtalir et al., 2018*) where initial dimerization of SMARCC (the mammalian Rsc8 homolog) is followed by the sequential binding of SMARCD (the mammalian Rsc6 homolog), SMARCB (the mammalian Sfh1 homolog), and SMARCE (no known RSC/yeast homolog) to form the BAF core.

Subsequent steps of RSC assembly likely involve binding of Rsc9 and Sth1. The armadillo repeat domain of Rsc9 binds the Rsc8 CC and Rsc6 to form most of the body lobe, while the scaffolding domains (I and II) of Sth1 contribute to both the head and body regions and exit towards the flexible leg lobe. The integration of Sth1 into the complex would also lead to recruitment of the components of the ARP module (Arp7, Arp9 and Rtt102). This order of events is derived from the corresponding phase in the assembly of mammalian SWI/SNF complexes (*Mashtalir et al., 2018*), which involves association of either ARID1 or ARID2 with the BAF core. ARID1 and ARID2 are paralogs, and both are orthologs to Rsc9, while the rest of the BAF complex-specific factors have no clear yeast orthologs. The incorporation of the ARID subunits is followed by the addition of the ATPase module, which includes the conserved SMARCA, ACTB, and ACTL6A subunits, which are the homologs of yeast Sth1, Arp7, and Arp9, respectively.

It is important to note that this initial assembly process primarily involves evolutionarily conserved subunits and domains and is thus likely conserved for all SWI/SNF chromatin remodelers (*Kadoch and Crabtree, 2015*). This model is supported by the structural similarities we observe between the yeast SWI/SNF and RSC complexes (and in agreement with a recently reported structure of yeast SWI/SNF bound to a nucleosome by *Han et al., 2019*). In the case of RSC, assembly

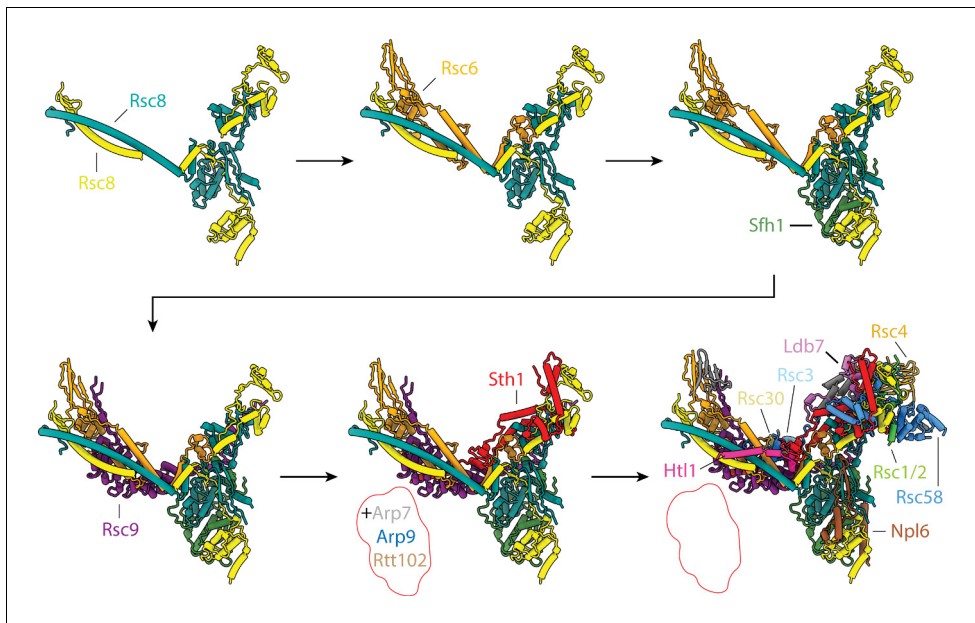

**Figure 7.** Proposed model of RSC assembly. Proposed assembly process of RSC highlighting the individual, initial stages, which involve only evolutionarily conserved subunits.

would then continue with the addition of the yeast specific factor Npl6, which completes the arm lobe, and RSC-specific subunits Htl1, Rsc58, Rsc1/2, Rsc4, and Ldb7. These subunits either contribute to the scaffold of the complex (Htl1 and Ldb7), serve to anchor bromodomains to the core (Rsc1/2 and Rsc4), or both (Rsc58). The last two subunits to be recruited to the complex are likely the RSC-specific subunits Rsc3 and Rsc30, which form the tail module.

## Conclusions

Our studies provide the detailed structure for the conserved core of SWI/SNF complexes and lead to a model of assembly for this family of remodelers that agrees with previous biochemical data on mammalian complexes. Our work also provides a model of how RSC stably contacts the core histones in a nucleosome via its body module, while engaging nucleosomal DNA via the Sth1 ATPase. Both the structural integrity of the core and the interaction with nucleosomes rely on evolutionarily conserved subunits, while the non-conserved proteins in RSC are likely to play a role in recruitment and regulation of the complex. Future studies will be needed to understand the mechanisms by which individual members of the SWI/SNF family of chromatin remodelers are brought to different genome loci and are distinctly regulated.

## Materials and methods

### Protein purification

SWI/SNF was purified from *Saccharomyces cerivisiae* using a modified TAP purification as described in *Nagai et al. (2017)*; *Puig et al. (2001)*. A strain modified with a TAP tag on the C terminus of SNF2 was obtained from GE Dharmacon and grown at 30℃ in YPD. 20 L of cells were harvested at OD six and lysed using a cryo-mill. The ground cells were resuspended in a lysis buffer (50 mM HEPES pH 7.9, 25 mM ammonium sulfate, 0.5 mM EDTA, 100 µM zinc sulfate, 5% glycerol, 5 mM DTT, 10 µM leupeptin, protease inhibitor,. 01% NP-40) and dounced to ensure homogenization. The lysate was spun at 11,000 g for 20 min. The supernatant was removed and brought to 200 mM ammonium sulfate, followed by an addition of polyethyleneimine (0.2% final concentration) to precipitate DNA. The sample was spun again at 17,000 g for 40 min. Supernatant was removed and brought up to 2.2 M ammonium sulfate, then spun again at 17,000 g for 40 min. The protein pellet was resuspended in buffer (lysis buffer with 0 M ammonium sulfate and 2 mM DTT) and the

ammonium sulfate concentration was brought up to 400 mM. IgG resin (1 ml packed) was equilibrated and incubated for 4 hr. Resin was washed with 500 mM ammonium sulfate buffer, then 1x with 100 mM ammonium sulfate buffer. Protein was released via TEV cleavage by overnight incubation with 20 µg TEV (MacroLab) in AS-100. Following cleavage, samples were spun to collect supernatant. The $CaCl_2$ concentration of the supernatant was brought up to 2 mM and loaded onto a 100 µl equilibrated CBP resin. After 4 hr of incubation, supernatant was removed and the resin was washed 4x (20 mM HEPES pH 7.9, 2 mM $MgCl_2$, 10% glycerol, 250 mM KCl, 50 µM $ZnCl_2$, 2 mM $CaCl_2$, 10 µM leupeptin, 1 mM TCEP, 0.01% NP-40). The sample was eluted with one volume elution buffer (20 mM HEPES pH 7.9, 2 mM $MgCl_2$, 10% glycerol, 250 mM KCl, 50 µM $ZnCl_2$, 2 mM EGTA, 10 µM leupeptin, 1 mM TCEP, 0.01% NP-40) for 30 min, then 15 min for each subsequent elution. Samples were aliquoted, frozen in liquid nitrogen and stored at −80℃.

RSC was purified from *Saccharomyces cerivisiae* using a TAP-tag method as described (*Puig et al., 2001*; *Dutta et al., 2017*). A strain modified with a TAP tag at the C terminus of STH1 was obtained from GE Dharmacon and grown at 30℃ in YPD. 10 L of cells were harvested at OD seven and lysed using a cryo-mill. The ground cells were resuspended in a lysis buffer (50 mM HEPES pH 7.9, 250 mM KCl, 0.5 mM EDTA, 100 µM $ZnSO_4$, 10% glycerol, 2 mM DTT, 10 µM leupeptin, protease inhibitor, 0.05% NP-40) and dounced to ensure homogenization, followed by an addition of 18 µL of benzonase (Sigma) while spinning on ice. After 10 min, another 18 µL of benzonase were added. Heparin (Sigma) was then added slowly to a final concentration of 0.5 mg/mL and incubated for 10 min. The lysate was then spun for 90 min at 17,000 g. Supernatant was collected and clarified through a column frit, then bound to 2.5 mL packed IgG resin and incubated for 4 hr at 4℃. After incubation, the supernatant was removed. TCEP was added to the lysis buffer for a final concentration of 0.5 mM and washed 5x over the resin. The sample was released via overnight incubation with 25 µg TEV protease in one volume buffer. Following cleavage, samples were spun to collect supernatant. The $CaCl_2$ concentration of the supernatant was brought up to 2 mM and loaded onto 100 µL CBP resin equilibrated five times with wash buffer (20 mM HEPES pH 7.9, 10% glycerol, 150 mM KCl, 50 µM $ZnCl_2$, 2 mM $CaCl_2$, 10 µM leupeptin, 1 mM TCEP, 0.01% NP-40). After 4 hr of incubation, the supernatant was removed and the resin was washed 2x with wash buffer with 750 mM KCl, then 3x with wash buffer with 250 mM KCl buffer, then 1x with wash buffer without $CaCl_2$. The sample was eluted with one volume (20 mM HEPES pH 7.9, 2 mM $MgCl_2$, 10% glycerol, 150 mM KCl, 50 µM ZnCl, 2 mM EGTA, 10 µM leupeptin, 1 mM TCEP, 0.01% NP-40) for 30 min, then 15 min for each subsequent elution. Samples were aliquoted, frozen in liquid nitrogen and stored at −80℃.

## Chemical crosslinking mass spectrometry

We used the ~250 µg RSC2 TAP-tag purified RSC complex and crosslinked by 4 mM bis(sulfosuccinimidyl)suberate (BS3) at RT for 2 hr. Sample processing and mass spectrometry data analyses were done as described before (*Mashtalir et al., 2018*). After plink2 and Nexus database searches against the RSC subunit sequences, about 6.6% of interlinked spectra and 3.1% intralinked spectra were removed after manually spectrum checking. All the crosslinked spectra can be viewed at https://www.yeastrc.org/proxl_public/viewProject.do?project_id=234.

## Negative stain sample preparation and data processing

For negative stain, the samples were cross-linked at room temperature for 5 min using 1 mM final concentration of BS3. After cross-linking, 4 µL were applied to a glow discharged continuous carbon grid for 5 min, then stained with uranyl formate. A tilted negative stain data set was collected on a Tecnai F20 microscope (FEI) operated at 120 keV and equipped with an Ultrascan 4000 camera (Gatan). Data were collected using Leginon data acquisition software (*Suloway et al., 2005*). The CTF parameters were estimated using Gctf (version 1.16) and particles were picked using Gautomatch (version 0.50, from K Zhang, MRC-LMB, Cambridge) using gaussian blob templates (*Zhang, 2016*). Data processing was done using Relion (version 3.0) (*Zivanov et al., 2018*). The negative stain structure (EMD-6834) from *Zhang et al. (2018)* was used as an initial model. Extracted particles were subjected to 2D classification and 3D classification to obtain a homogenous population. Particles that went into the best classes were then refined.

## Cryo-EM sample preparation

For cryo-EM sample preparation we used a Vitrobot Mark IV (FEI). RSC was crosslinked on ice using 1 mM BS3 (Thermo Fisher Scientific) for 15 min before 4 µL of sample was applied to either a 2/2 holey carbon or 1.2/1.3 UltrAuFoil grids (Quantifoil) at 4°C under 100% humidity. Grids were cleaned using Toguro plasma cleaner, using the 3 nm carbon setting run twice. The sample was immediately blotted away using Whatman #1 for 2–3 s at 0 N force and then immediately plunge frozen in liquid ethane cooled by liquid nitrogen. For the RSC-nucleosome complex sample preparation, 4 µL of RSC (2 pmol) and 1 µL of nucleosome (H3K4me3 and H3K(9/14/18)ac) (1 pmol) (Epicypher) were incubated for 10 min at 30 °C followed by the addition of 0.5 µL of AMPPNP (0.5 nmol) and an additional incubation for 10 min at 30°C. The samples were then placed on ice and crosslinked with 1 mM BS3 for 15 min. RSC-nucleosome samples were prepared in the same way as RSC.

## Cryo-EM data collection

For the RSC sample, frozen grids were clipped and transferred to the autoloader of a Titan Krios electron microscope (Thermo Fischer Scientific) operating at 300 keV (PNCC). Images were recorded with a K3 direct electron detector (Gatan) operating in super-resolution mode at a calibrated magnification of 46,339 (1.079 Å/pixel) and a mean defocus of −1.04 µm with a 0.24 µm standard deviation, using the SerialEM data collection software (*Schorb et al., 2019*). 50-frame exposures were taken at 0.06 s per frame, using a dose rate of 11.455 e⁻/pixel/s (0.6 e⁻ $Å^{-2}$ per frame), corresponding to a total dose of 40 e⁻$Å^{-2}$ per micrograph (*Figure 1—figure supplement 3*). A total of 8122 movies were collected from a total of 3 grids.

For the RSC-nucleosome sample, frozen grids were clipped and transferred to the autoloader of a Talos Arctica electron microscope (Thermo Fischer Scientific) operating at 200 keV acceleration voltage (UCB). Images were recorded with a K3 direct electron detector (Gatan) operating in super-resolution mode at a calibrated magnification of 43,859 (1.14 Å/pixel) and a mean defocus of −1.66 µm with a 0.41 µm standard deviation, using the SerialEM data collection software (*Schorb et al., 2019*). 50-frame exposures were taken at 0.065 s per frame, using a dose rate of 11.838 e⁻/pixel/s (1 e⁻ Å (*Yen et al., 2012*) per frame), corresponding to a total dose of 50 e⁻$Å^{-2}$ per micrograph (*Figure 1—figure supplement 9*). A total of 9190 movie were collected from a single grid.

## Cryo-EM data processing

All data processing was performed using Relion3 (version 3.0) (*Zivanov et al., 2018*). For the RSC dataset, whole movie frames were aligned and binned by 2 (1.079 Å/pixel) with MotionCor2 to correct for specimen motion The CTF parameters were estimated using Gctf (*Zhang, 2016*; *Zheng et al., 2017*). 1,245,498 particles were picked with LoG picker. Particles were extracted binned by 4 (4.316 Å/pixel) and subjected to two- and three-dimensional classification to remove ice and empty picks, which resulted in 1,074,750 particles. The negative stain reconstruction was used as an initial model for 3D classification. The particles were then centered and reextracted bin 1.33 (1.4386 Å/pixel) and refined. The refinement was performed without a mask and resulted in a reconstruction were only the core of the complex was well resolved. The refinement was then continued with a mask around the core. The masked refinement resulted in a 3.8 Å-resolution map of the core. Masked local search 3D classification was performed to select for the best particles. The best class, containing 252,918 particles, was selected and refined, resulting in a 3.4 Å map. Three iterations of CTF refinement, particle polishing and 3D refinement were performed, which resulted in reconstructions at 3.26, 3.21 and 3.18 Å (*Zivanov et al., 2019*). Local search 3D classification was performed to select the best particles. The best class, containing 192,066 particle images, was selected and further refined, resulting in a 3.14 Å-resolution map. One last iteration of CTF refinement, particle polishing and refinement was performed and led to a reconstructions at 3.07 Å resolution. The core was then subjected to multibody refinement by masking each of the three lobes separately (*Nakane et al., 2018*). The arm, head and body refined to 3.23, 3.14, and 2.96 Å respectively. 3D classification of the partially signal subtracted particles was then performed. For the arm lobe a single good class was identified, which when refined resulted in a 3.16 Å map. For the head and body lobes two good classes were found, with one containing several extra helices. The classes containing the extra density were selected and refined. The head lobe refined to 3.07 Å, and the body refined to 3.48 Å. To characterize the two flexible lobes of the complex 3D classification was performed for

the 192,066 subset of particles. The classification with four classes resulted in a continuum of states for the leg lobe. The complete tail lobe was only present in two of the four classes, with the tail in two different conformations.

For the RSC dataset, whole movie frames were aligned and binned by 2 (1.14 Å/pixel) with MotionCor2 to correct for specimen motion The CTF parameters were estimated using Gctf (*Zhang, 2016*; *Zheng et al., 2017*). 2,327,957 particles were picked with LoG picker. Particles were extracted binned by 4 (4.56 Å/pixel) and subjected to two- and three-dimensional classification to remove ice, empty picks, free nucleosomes which resulted in 48,222 particles. The cryo-EM reconstruction of RSC alone was used as an initial model for 3D classification. The particles were refined to 19 Å and was subjected to another round to 3D classification. Further refinement from 3D classification did not improve reconstruction quality (not shown).

Some of the software packages mentioned above were configured by SBgrid (*Morin et al., 2013*).

## Model building and refinement

The model for the RSC core was generated by manually building a poly-alanine trace through the final global refinement and multibody maps in COOT (*Emsley et al., 2010*). Each of the chains was then identified with the help of blobMapper.py (*Zukin, 2020*; copy archived at https://github.com/elifesciences-publications/blobMapper) and secondary structure predication (*Figure 1—figure supplement 4*) (*Jones, 1999*; *Buchan et al., 2013*; *Jones and Cozzetto, 2015*). The resulting coordinate model was iteratively refined using the real space refinement algorithm implemented in PHENIX (*Afonine, 2018a*). Ramachandran, secondary structure, $C_\beta$, and rotamer restraints, as well as bond length and bond angle restraints for the $Zn^{2+}$ ion in the Rsc8 ZZ domain, were used throughout to ensure good model geometry. The final round of refinement comprised 5 rounds of global minimization as well as b-factor refinement and used a resolution limit of 3.1 Å, according to the average resolution of cryo-EM maps used (*Figure 1—figure supplement 7F*) in order to avoid overfitting of the model. The refinement weight was automatically determined by PHENIX, which monitors the bond length and bond angle deviations to maintain good model geometry and avoids over-fitting of the model to the map (*Afonine et al., 2018b*). The model was validated using MTRIAGE and MOLPROBITY within PHENIX (*Afonine et al., 2018b*). The refinement statistics are given in (*Figure 1—figure supplement 7H*) and show values typical for structures in this resolution range (MOLPROBITY score = 2.1) (*Duan and Smerdon, 2014*). The FSC curve between the model and the map shows good correlation up to 3.2 Å resolution according to the FSC = 0.5 criterion (*Figure 1—figure supplement 7G*) (*Afonine et al., 2018b*).

The model of the RSC-NCP complex used for visualization was generated by docking our model of the RSC core, the crystal structure of the Arp module (PDB 4I6M: Arp7, Arp9, Rtt102 and Snf2-HSA) (*Schubert et al., 2013*) and cryo-EM structure of Snf2-MMTV nucleosome complex bound at the SHL2 with ADP (PDB 6IY2: Snf2, nucleosome) (*Li et al., 2019*). The sequence for the Snf2-HSA helix in 4I6M and Snf2 helicase domain in 6IY2 were aligned to the Sth1 sequence and mutated in COOT (*Emsley et al., 2010*). For deposition of the coordinate model to the PDB, we replaced the model of the Snf2-MMTV nucleosome complex with the structure of a human nucleosome (PDB 2CV5) (*Tsunaka et al., 2005*), without bound ATPase domains, because the ATPase domain of Sth1 was not resolved in our cryo-EM map.

Some of the software packages mentioned above were configured by SBgrid (*Morin et al., 2013*).

## Creation of figures and movies

Depiction of molecular models were generated using PyMOL (The PyMOL Molecular Graphics System, version 1.8, Schrödinger), the UCSF Chimera package from the Computer Graphics Laboratory, University of California, San Francisco (supported by National Institutes of Health P41 RR-01081) and UCSF ChimeraX developed by the Resource for Biocomputing, Visualization, and Informatics at the University of California, San Francisco, with support from National Institutes of Health R01-GM129325 and the Office of Cyber Infrastructure and Computational Biology, National Institute of Allergy and Infectious Diseases (*Pettersen et al., 2004*; *Goddard et al., 2018*). Protein domains

graphs (*Figure 1—figure supplement 4*; *Figure 4*) were generated using domainsGraph.py (*Patel, 2020*; copy archived at https://github.com/elifesciences-publications/domainsGraph).

Some of the software packages mentioned above were configured by SBgrid (*Morin et al., 2013*).

## Acknowledgements

We thank S Baskaranm, H Asahara, and R Lesch for assistance with yeast work, A Iavarone for performing in-gel mass spectrometry data collection and analysis, P Grob, and D Toso for electron microscopy support, A Chintangal and P Tobias for computing support, and C Yoshioka and the OSHU Cryo-EM Facility for help with data collection. A portion of this research was supported by NIH grant U24GM129547 and performed at the PNCC at OHSU and accessed through EMSL (grid.436923.9), a DOE Office of Science User Facility sponsored by the Office of Biological and Environmental Research. This work was funded through NIGMS grants R01-GM63072 and R35-GM127018 to EN, and R01-GM110064 and R01-HL133678 to JR. EN is a Howard Hughes Medical Institute Investigator.

## Additional information

### Funding

| Funder | Grant reference number | Author |
| --- | --- | --- |
| National Institutes of Health | U24GM129547 | Eva Nogales |
| National Institute of General Medical Sciences | R01-GM63072 | Eva Nogales |
| National Institute of General Medical Sciences | R35-GM127018 | Eva Nogales |
| National Institute of General Medical Sciences | R01-GM110064 | Jeff Ranish |
| National Institute of General Medical Sciences | R01-HL133678 | Jeff Ranish |
| Howard Hughes Medical Institute | | Eva Nogales |

The funders had no role in study design, data collection and interpretation, or the decision to submit the work for publication.

### Author contributions

Avinash B Patel, Conceptualization, Investigation, Visualization, Writing - original draft, Writing - review and editing, Prepared, collected and processed EM data, Built atomic model; Camille M Moore, Investigation, Visualization, Writing - original draft, Prepared, collected and processed EM data; Basil J Greber, Investigation, Writing - review and editing, Built atomic model and refined the atomic coordinate model; Jie Luo, Investigation, Writing - review and editing, Performed cross-linking mass spectrometry analysis; Stefan A Zukin, Software, Wrote blobMapper.py program; Jeff Ranish, Eva Nogales, Funding acquisition, Writing - review and editing

### Author ORCIDs

Avinash B Patel (iD) https://orcid.org/0000-0001-9140-8375
Basil J Greber (iD) http://orcid.org/0000-0001-9379-7159
Eva Nogales (iD) https://orcid.org/0000-0001-9816-3681

### Decision letter and Author response

Decision letter https://doi.org/10.7554/eLife.54449.sa1
Author response https://doi.org/10.7554/eLife.54449.sa2

## Additional files

### Supplementary files

• Transparent reporting form

### Data availability

The cryo-EM maps and coordinate models have been deposited in the Electron Microscopy Data Bank with the accession codes EMD-21107 (RSC core), EMD-21105 (head lobe multibody), EMD-21103 (body lobe multibody), EMD-21098 (arm lobe multibody), EMD-21106 (head lobe classified), EMD-21102 (body lobe classified), EMD-21104 (arm lobe classified), EMD-21114 (RSC-NCP locked) and EMD-21110 (RSC-NCP swiveled) and in the Protein Data Bank with the accession codes PDB-6V8O (RSC core) and PDB-6V92 (RSC-NCP).

The following datasets were generated:

| Author(s) | Year | Dataset title | Dataset URL | Database and Identifier |
|---|---|---|---|---|
| Patel AB, Moore CM, Greber BJ, Nogales E | 2019 | RSC core | https://www.ebi.ac.uk/pdbe/entry/emdb/EMD-21107 | Electron Microscopy Data Bank, EMD-21107 |
| Patel AB, Moore CM, Greber BJ, Nogales E | 2019 | head lobe multibody | https://www.ebi.ac.uk/pdbe/entry/emdb/EMD-21105 | Electron Microscopy Data Bank, EMD-21105 |
| Patel AB, Moore CM, Greber BJ, Nogales E | 2019 | body lobe multibody | https://www.ebi.ac.uk/pdbe/entry/emdb/EMD-21103 | Electron Microscopy Data Bank, EMD-21103 |
| Patel AB, Moore CM, Greber BJ, Nogales E | 2019 | arm lobe multibody | https://www.ebi.ac.uk/pdbe/entry/emdb/EMD-21098 | Electron Microscopy Data Bank, EMD-21098 |
| Patel AB, Moore CM, Greber BJ, Nogales E | 2019 | head lobe classified | https://www.ebi.ac.uk/pdbe/entry/emdb/EMD-21106 | Electron Microscopy Data Bank, EMD-21106 |
| Patel AB, Moore CM, Greber BJ, Nogales E | 2019 | body lobe classified | https://www.ebi.ac.uk/pdbe/entry/emdb/EMD-21102 | Electron Microscopy Data Bank, EMD-21102 |
| Patel AB, Moore CM, Greber BJ, Nogales E | 2019 | arm lobe classified | https://www.ebi.ac.uk/pdbe/entry/emdb/EMD-21104 | Electron Microscopy Data Bank, EMD-21104 |
| Patel AB, Moore CM, Greber BJ, Nogales E | 2019 | RSC-NCP locked | https://www.ebi.ac.uk/pdbe/entry/emdb/EMD-21114 | Electron Microscopy Data Bank, EMD-21114 |
| Patel AB, Moore CM, Greber BJ, Nogales E | 2019 | RSC-NCP swiveled | https://www.ebi.ac.uk/pdbe/entry/emdb/EMD-21110 | Electron Microscopy Data Bank, EMD-21110 |
| Patel AB, Moore CM, Greber BJ, Nogales E | 2019 | RSC core | https://www.rcsb.org/structure/6V8O | RCSB Protein Data Bank, 6V8O |
| Patel AB, Moore CM, Greber BJ, Nogales E | 2019 | RSC-NCP | https://www.rcsb.org/structure/6V92 | RCSB Protein Data Bank, 6V92 |

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

## Appendix 1

### Structural comparison of RSC complexes

During the submission of this manuscript, another structure of RSC bound to a nucleosome was published by *Ye et al. (2019)* and *Wagner et al. (2019)*. The structures by Ye et al and our own are very similar with few exceptions. These mainly stem from the apparent dissociation of parts of Ldb7 from the body and the binding of the ATPase domain to nucleosome in the reconstruction of Ye et al (*Appendix 1—figure 1*). The structure by Wagner et al has not yet been deposited so direct comparison could not be made but the overall structure appears very similar.

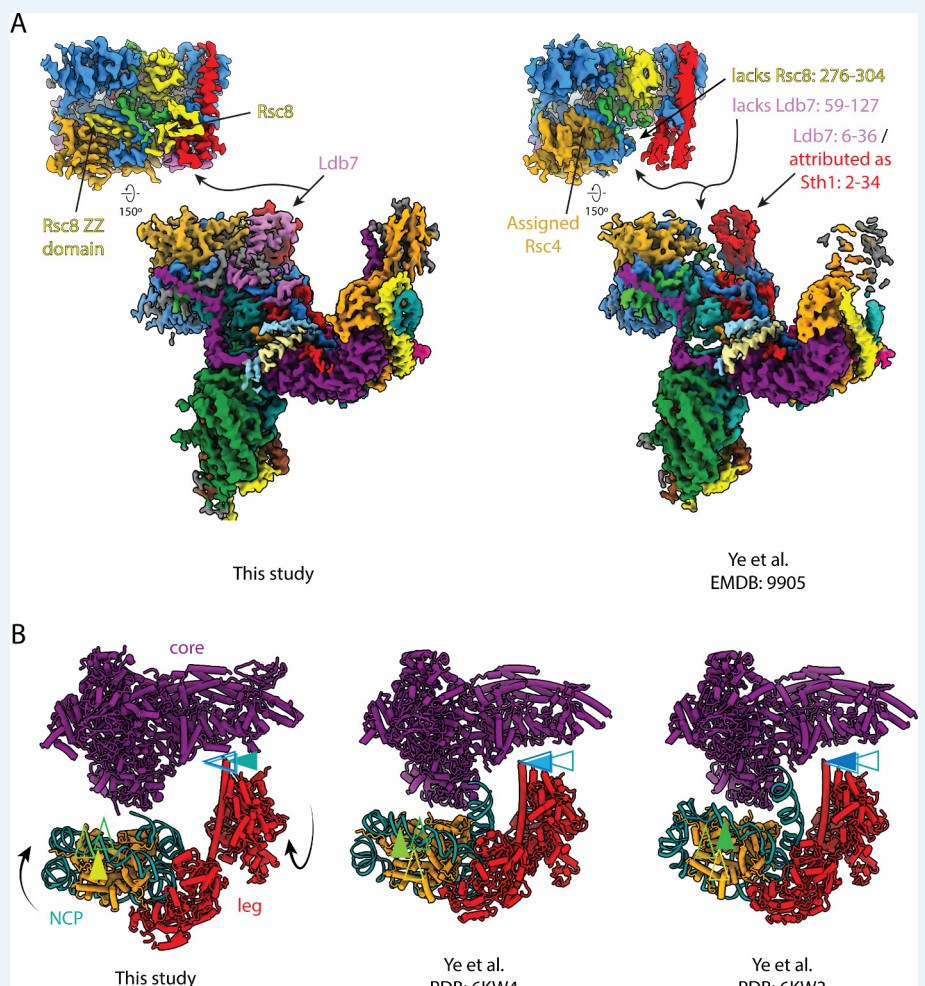

**Appendix 1—figure 1.** Structural comparison of RSC complexes. (**A**) Structures of the RSC core from this study (left) and (*Ye et al., 2019*) (right) . (**B**) Structures of the RSC core from this study (left) and (*Ye et al., 2019*) (middle and right). Arrows indicate relative positions of nucleosome super-helical location 0 (SHL0) (yellow-green) and HSA helix (teal-blue).
The online version of this article includes the following figure supplement(s) for figure app11:
**Appendix 1—Figure 1 supplement 1.** Interpretation of densities for Ldb7, Rsc8 and Rsc4.

 The region we attribute to be residues 59–127 of Ldb7 appears to be missing from the reconstruction of Ye et al (*Appendix 1—figure 1A*). Additionally, segments of some chains we observe around this site appear to have been more flexible in the Ye et al. structure. The region of Sth1 attributed to residues (2-34) by Ye et al is assigned by us to correspond to

Ldb7 residues 6–36 (*Appendix 1—figure 1 – Appendix 1—figure 1—figure supplement 1A*). Additionally, two β-strands within the β-sandwich that we identified as being the ZZ domain of Rsc8 were proposed to correspond to Rsc4 residues 429–449 (*Appendix 1—figure 1 – Appendix 1—figure 1—figure supplement 1B C, D*). This resulted in a change in register for a third β-strand (residues 364–390) (*Appendix 1—figure 1 – Appendix 1—figure 1—figure supplement 1E, F, G, H*). Within our map, we see connecting density between the SANT and ZZ domains of Rsc8 that allowed us to trace this chain.

Our docking of the ARP module into the leg density and prediction that the ATPase module would bind at the SHL2 site of the nucleosome matched well with the structure of Ye et al. Within their structure they were able to observe the binding of the ATPase domain to the SHL2 site. This likely explains the shift of the ARP module between our two reconstructions, with the ARP module closer to the nucleosome in their reconstruction then in ours. This likely means that the ARP module remains flexible to some degree until the ATPase module binds the nucleosome.

One major difference between the three structure of RSC bound to the nucleosome was the length of the DNA used in generating the nucleosome. We had only the 147 bp WIdom 601 sequence with no extranucleosomal DNA, Ye et al used the 147 bp WIdom 601 sequence with an additional 20 bp on one end making the total length 174 bp and Wagner et al used the 145 bp WIdom 601 with an additional 58 bp on one end and 37 bp on the other making the total length 237 bp. Due to the extra length of the extranucleosomal DNA in their structure, Wagner et al were able to observe the extranucleosomal DNA extending to the tail lobe where it appears to bind. In our RSC-NCP sample we also observe the tail lobe binding DNA but, in that case, the entire tail lobe is swiveled down towards the nucleosome and is interacting with the nucleosomal DNA. Due to the lack of resolution in both structures for this region, it was not possible to determine structure of the tail lobe and therefore, future studies will be required to identify full function of the tail of RSC.

**Appendix 2**

## Sfh1 and SMARCB1 C-terminal α-helix binds nucleosomal acidic patch

During the submission of this manuscript two studies by *Wagner et al. (2019)* and *Valencia et al. (2019)* showed that the the C-terminal region of Sfh1 and SMARCB1, respectively, interacts with the acidic patch of the nucleosome. In the study by Wagner et al, the structure of RSC-NCP was determined using cryo-EM and it was shown that a portion of the C-terminus of Sfh1 forms an α-helix and binds the acidic patch of the nucleosome. Valencia et al show that the C-terminal region of SMARCB1 (Sfh1 homolog) forms an α-helix by NMR. Additionally, these authors showed that this α-helix will crosslink to residues near the acidic patch of the nucleosome and by computational docking that this α-helix will bind the acidic patch of the nucleosome. Valencia et al also showed that deletions of mutations in this helix reduced remodeling efficiency in vitro but did not prevent nuclear localization in vivo. These last two findings support our proposal that that Sfh1 and the arm lobe plat a role in nucleosome engagement but not RSC recruitment.

## Appendix 3

### Structural comparison of RSC and SWI/SNF

During the submission of this manuscript the structure of SWI/SNF bound to a nucleosome was published by *Han et al. (2019)*. While at the time of final submission of this manuscript, the structure by Han et al has not been deposited yet, and a direct comparison could not be made, several clear similarities and differences are apparent. While the overall structure of SWI/SNF and RSC appear to be similar with respect to organization of homologous subunits, the resulting shape of the complex differs. The core of RSC appears to have three distinct lobes (head, body and arm) but in the case of SWI/SNF the core appears to form a single triangular shaped lobe.

The biggest difference between the two remodelers appears to stem form the lack of many of the head lobe components of RSC in SWI/SNF (e.g. Rsc58, Rsc1/2, Rsc4 and Ldb7). Because of this, the remaining head lobe components of SWI/SNF, the SANT domains of Swi3 (Rsc8 homolog) and the anchor domain of Snf2 (homologous to the Scaffold II domain of Sth1), interact more closely with the body lobe.

The body lobe of SWI/SNF is organized in a similar way as in RSC, but the overall shape appears different. While Swi1, Swi3 and Swp73 are similar to their RSC counterparts Rsc9, Rsc8 and Rsc6, they are different enough so that the body appears to curve toward the arm lobe in SWI/SNF as opposed to away from it, as in RSC. Interestingly it appears that Swi6 may be homologous to Htl1 as they share similar binding sites in the helical bundle within the body.

The region most similar between RSC and SWI/SNF appears to be the arm lobe. This may be due to structural or functional constraints, considering that it is the arm lobe of both complexes that interacts the nucleosome. However, the arms of the two complexes bind over the nucleosome at different positions. The arm of RSC binds over the H2A/H2B dimer of the nucleosome while in SWI/SNF, the arm binds over the H3/H4 surface. So, it remains to be seen why the binding sites for the arm lobes is different between these two complexes despite the lobes themselves being so similar.

