## [Decision Letter]

**Acceptance summary:**

The RSC complex is nucleosome remodeler that maintains nucleosome-free regions in yeast and is the most abundant member of the SWI/SNF family of enzymes. This paper reports the cryo EM structure of the RSC complex core, containing 14 of the 17 RSC subunits. The structure reveals the architecture of the complex, and the similarities and differences as compared to other SWI/SNF remodelers. A low-resolution structure of RSC bound to a nucleosome core particle suggest a model for how RSC engages its substrate. Taken together, this work is an important advance in the chromatin and transcription fields, and provides a structural basis for interpreting previous biochemical studies in order to derive a possible model for RSC assembly and function.

---

## [Author Response]

[Editors' note: we include below the reviews that the authors received from another journal, along with the authors’ responses.]

As you will see the paper has been significantly changed. While all the original content is still present, we have restructured most of this. This was in part to fit the *eLife* format but also to accommodate the additional discussion on analyzing our structure. In addition to these changes and additions we have also added (as was asked) a comparison of our structure with the one published in Science by Ye et al. The comparison has been added as Appendix I. We also included two other Appendices (II and III) which cover three new papers: 1) [Valencia et al., Cell] on the analysis of SMARCB1 C-terminus and its ability to bind the acidic patch of the nucleosome, 2) [Wagner et al., bioRxiv] on the structure of RSC bound to a nucleosome, and 3) on the SWI/SNF bound to a nucleosome.

Overall, we felt that while some of the issues raised by the reviewers are understandable and will help us revise our manuscript, others are not.

One of the major areas of concern raised was that the assembly process could not be determined from our structure alone. This is indeed true, but it misses the fact that our model is based not only on our structure, but also on published literature (citation 16: Mashtalir, N. et al.) that provides biochemical data in support of our model. To emphasize this, we will change “The pattern of protein‐protein interactions observed in our structure provides insight into RSC assembly” to “Based on the observed protein-protein contacts in our structure of RSC and previous mass spectrometry analysis of the assembly of human BAF complexes, we propose a model for RSC assembly”. In this way we clarify that the model we propose originates from previous biochemical data and that our structure can now help to rationalize this model at the domain level. While we thought that this would be obvious, given that the literature describing the assembly process is highly cited in our work, it appears that both reviewers 1 and 2 did not recognize this and we will make it as clear as possible in the revised manuscript.

We have now included more extensive discussion on this by referencing the key intermediates that Mashtalir et al. have found.

The other major issue that was brought up (by reviewer 1 and 2) was that our cryo-EM structure of the RSC-NCP was too low of a resolution and that our claims about the structure were too speculative. While it is true that our structure of nucleosome-bound RSC is at low resolution (similar to previously published structures) nearly all previous reconstructions were of very low quality (e.g. Leschziner et al., 2007, Yuriy et al., NSMB 2008, Dechassa et al., 2008 and Zhang et al., 2018). It should be noted that resolution does not always accurately measure map quality. Therefore, while our map is at a similar resolution to previous reconstructions it does not suffer from the clear misalignment, anisotropy, and apparent flattening of previous works. Specifically, the EM map by Zhang et al., which was suggested by reviewer 1 to be of similar quality to ours, suffers from severe flattening and precludes fitting of the molecular components, as revealed by even the most cursory inspection. We therefore strongly disagree with the claim that our structure does not provide significant new insight into the architecture of nucleosome-bound RSC. Based on our map, we can unambiguously dock and assign all observable density, the core, the ARP module and the nucleosome. As far as not being able to observe any bromodomain density we would like to point out that one would not expect to see these domains because they are flexibly attached to the core of the complex and even in the case of the nucleosome bound complex, they would be interacting with histone tails, which are themselves also flexible.

The remaining issues raised by reviewer 1 are hard to comprehend.

Specifically, (i) the issue with Rsc3 and Rsc30 being sub-stoichiometric has been well documented and the corresponding literature has been cited in our paper; (ii) we do not think that measuring complex stoichiometry based on the gel or running a glycerol gradient to ascertain that we indeed purified an assembled complex is worthwhile, given that we have a 3-Å resolution structure of the core of complex; (iii) we cite 2 of the 3 key papers the reviewer mentions we “missed” (which includes one authored by some of the co-authors of our study), while the third is only tangentially relevant.

It is our assessment that the paper can be revised to clarify some of the issues raised by the reviewers. However, several of the issues raised are not applicable, as detailed above.

In addition to these general considerations, we provide point-by-point answers to the reviewers’ comments below.

Reviewer #1:

In the first part of this paper, the authors use cryo-EM to determine the structure of free RSC, one of two SWI/SNF complexes present in yeast, and were able to resolve the core of this complex to 3.1 angstroms. The core has a more rigid region consisting of a head, body, and arm regions plus a more flexible leg region. The structure they observe is at a higher resolution than any previously reported structure for a SWI/SNF complex and provides detailed information about the subunit-subunit interactions within this complex. They complement their cryo-EM structure by also probing the structure using chemical crosslinking and mass spectrometry (CL^-^MS) in collaboration with the Ranish lab.In support of their CL^-^MS results they cite similar data from the Kadoch lab, but missed the earlier work from Sen et al., 2017, using yeast SWI/SNF that is actually more similar to RSC.

We do not use the work of the Kadoch lab to support our CL^-^MS results. Their work is mainly cited for the assembly process that they propose. The work of Sen et al. is cited (number 22) for other findings.

From their data it was not clear how they handled the heterogeneity of RSC since their purified enzyme had both Rsc1 or Rsc2, which are mutually exclusively of each other in the complex, and raises the question why they didn’t tag Rsc1 or Rsc2 so they could purify one or the other form of the RSC complex.

In our cryo-EM data processing we did not find that the complexes could be distinguished based on whether they contain Rsc1 or Rsc2. However, we did find a region in our cryo-EM map where we could model a small region of Rsc1 and 2 that is highly similar between these two proteins. It is worth noting that the sequences of Rsc1 and Rsc2 are >40% identical and another 20% is highly homologous, suggesting that they form highly similar contacts within RSC.

We reasoned that if Rsc1 or 2 lead to two structurally distinct complexes we would be able to distinguish these computationally, while otherwise, the two can be treated as the same. The latter ended up being the case as the regions that anchor the two proteins into the core of the complex are highly similar.

We have included an additional figure (Figure 1 - figure supplement 6) showing an alignment of the regions of RSC1 and 2 we modeled and its fit into the density.

It would be helpful if in the paper the authors stated which portions of each of the 14 different subunits were visualized in the core and were well resolved and which portions were too flexible to be resolved.Figure 1—figure supplement 4 contains this information.Figure 1—figure supplement 4 was essentially indiscernible and needs to be remade so that it can be actually read.

We have made a new version of the domain map figure for the main text (Figure 1F). It does not have the secondary structure prediction or sequence conservation mapped on to the domain map. Instead it the modeled region shown in much larger font. We chose to keep the original figure (we made some changes) in the supplement (Figure 1—figure supplement 4) because the secondary structure predication was used when building to assign proteins to chains. Additoinal we have included Figure 1 - figure supplement 5 to show the fit of various modeled regions into the map.

The purity and overall subunit stoichiometry in the purified RSC appear to be in question and was difficult to discern in the SDS-PAGE shown in Figure 1—figure supplement 1A. What type of gel was used, what was the percentage acrylamide, and what type of staining was used to visualize the proteins?

We used a BioRad 4-20% gel and stained with flamingo stain. We can rerun the gel to better separate the region that contains Rsc4, Rsc6, Rsc8, Rsc9, Npl6, Sfh1, Rsc58, Arp7, Arp9, but as the molecular weights differ only 20 kDa between the largest and smallest subunit of this group, complete separation of these bands is not likely.

Based on the in-solution tryptic digest and mass spectrometry data the sample is pure (Figure 1—figure supplement 2). The relative stoichiometry is difficult to determine for this complex using gel densitometry as many of the bands overlap in molecular weight.

If the gel was stained with Coomassie Blue then the authors need to give estimates of subunit stoichiometry based on staining intensity. Could the authors also comment on why the Rsc3 and Rsc30 subunits appear to be absent in 75% of their complexes and if this agrees with their stained gel?

The gel is not Coomassie Blue stained.

The fact that Rsc3 and 30 are substoichiometric has previously been reported, see citation 20 (El-Gebali et al., 2019).

Given the potential heterogeneity it is imperative to show how well assembled their complex is using such approaches as gel filtration or glycerol gradient at the minimum.

We can include negative stain micrographs and 2D class averages (Figure 1 -figure supplement 2) that show that we do not observe any partially assembled complexes. We also want to note that (i) we used complexes purified from a native source, making mis-assembly of the complex unlikely and (ii) we present in this paper a high-resolution structure of the core of this complex, which of course supports the notion that we are looking at a well-assembled and pure molecular assembly.

The negative stain comparison of RSC vs SWI/SNF was a good addition for showing the lack of a tail lobe that likely contains those proteins or domains not shared between the two complexes. Could the tail lobe contain the Rsc3 and Rsc30 subunits?

Yes, we do think this is the case and have stated “we found that the presence or absence of Rsc3 and Rsc30 correlates well with the presence or absence of most of the tail lobe, making the Rsc3 and Rsc30 subunits the most likely constituents of this very flexible lobe (now in Figure 3).” The reviewer apparently missed this part of our manuscript.

The authors also did not cite the yeast SWI/SNF work from Craig Peterson’s lab which showed the importance of Swi3 for scaffolding of the complex (Yang et al., Nat. Struct. Mol. Biol. 14:540-547 [2007]), while only citing the later work with BAF155/170.

We have added this citation. However, this paper only looked at how the SANT domain of Swi3 effected the assembly of SWI/SNF, and so we felt that the paper did not strongly support the fact that all domains of Rsc8 were important in scaffolding RSC.

This paper makes significant progress in unraveling the subunit architecture of the free RSC complex, but I think it goes too far in suggesting the order of RSC assembly based solely on the structure. A fair amount of emphasis is placed on the order of assembly as it is discussed in two paragraphs and as stated is merely the authors’ proposal without any additional supporting evidence.

We can change “The pattern of protein‐protein interactions observed in our structure provides insight into RSC assembly” to “Based on the observable protein-protein contacts in our structure of RSC, along with previous mass spectrometry analysis of the assembly of human BAF complexes, we propose a model for RSC assembly.”

Our aim was to build off the model that was recently proposed in the Kadoch lab’s recent paper (citation 16: Mashtalir, N. et al.). What our work adds to theirs is moleuclar detail, as we can show which subunits are interacting, and suggest what the Kadoch model would look like in real molecular terms.

Like we stated above we have significantly expanded the discussion on this by referencing the key intermediates that Mashtalir et al. have found.

The authors then visualize the RSC-nucleosome structure at a resolution of 19 angstroms, which is not significantly better than a recent report of Zhang et al., 2018, for the SWI/SNF complex. At this resolution it is not possible to readily discern the structural features or subunit organization of how RSC interact with nucleosomes. The other serious issue with this structure is the lack of observable density of the ATPase domain where it should be engaging nucleosomes.

While our resolution is not significantly better than the work of Zhang et al., our reconstruction is far more homogeneous and isotropic, and thus allowed use to unambiguously dock the structure of the nucleosome. Additionally, we performed cryo-EM where we are able to observe the nucleic acid of the nucleosome far better then what is possible in negative stain, which was the method used by Zhang et al. We also want to emphasize that the map by Zhang et al. suffers from severe flattening while ours does not.

While we are not certain of why we do not observe clear density for the ATPase domain, one possibility is that the stability of the interaction is reduced in the absence of extranucleosomal DNA. However, we believe the fact that we do see engagement with the nucleosome by the core and tail of RSC has added value, and propose that our structure describes an intermediate in the process of nucleosome engagement.

These observations raise the question as to whether the RSC-nucleosome complex formed is active for remodeling, and there is unfortunately no biochemical data to show whether this is the case or not.

We do not claim that our complex is active for remodeling. Instead we are suggesting that what we are observing is an intermediate in the process of nucleosome binding by RSC (Figure 4B).

We have changed two key words to indicate the speculative nature of our model.

It also was not clear as to why these nucleosomes were modified with lysines acetylated at residues 9, 14 and 18 and methylated at residue 4 of histone H3. Should these modifications change the interactions of RSC with nucleosomes, and if so shouldn’t there be an accompanying control of unmodified nucleosomes?

We decided to use these nucleosome as the acetylation marks have been shown to interact with RSC and its domains in numerous studies (see citation 24, 25, and 26 – more citations can be added). Our complex is therefore more representative of a physiological substrate compared to an entirely unmodified nucleosome.

It seems there should be more dialog as to why these particular histone modifications and potential supporting experimental data.

We can add the following statement:

“In order to shed light on the interaction of RSC with its substrate, we obtained a 19Å resolution map of RSC bound to a nucleosome core particle (NCP) modified with H3K4me3 and H3K(9/14/18)ac (Figure 1D, E, F; Figure 1—figure supplement 9). We used acetylated nucleosome because is has been shown that RSC has a higher affinity for H3 acetylated nucleosomes” (citations 24, 25, and 26 can be used as well as others).

The assumption that the structure of free RSC remains the same when bound to nucleosomes as used in this paper is too much of an extrapolation and suffers from the lack of data. There seems to be too much prediction and modeling used in this part, all because of the lack of resolution of the nucleosome-bound structure. The authors are left to make correlations based on previous data and are unable to truly move the field forward in terms of how RSC interacts with nucleosomes.

We disagree with this statement. While we do not have high resolution for the core of RSC in our RSC-NCP structure, we can still see that the overall shape remains the same. We can fit the core of RSC, the ARP module and NCP unambiguously into our RSC-NCP map.

We did use existing structures of the ARP and NCP and our newly determined RSC core structure for docking, and, as we stated, we were able to *unambiguously* place them within the lower resolution structure. For this reason, our model cannot consider “too much of an extrapolation.”

Lastly while we do not have high resolution for the regions that interact with the nucleosome, through the unambiguous docking of high-resolution structures of the pieces we are able to indicate what regions are interacting with the nucleosome (Sfh1, Rtt102 and Npl6) and which are not (the SWIRM, ZZ and SANT domains of Rsc8 – all of which have been predicted to bind DNA or nucleosomes).

Although the nucleosome has four acetylated lysines, there doesn’t seem to be any clear discernible structure showing the six bromodomains of RSC interacting with their likely targets. The authors also missed yet another earlier work showing the importance of RSC recruitment with acetylated histone H3 tails (see Chatterjee et al., 2011). I also found Figure 3 hard to read as the text is too small.

As we explain in the paper, the bromodomains are likely flexible with respect to the core. This is part of the reason we do not see them. We are also unlikely to see the bromodomains in the RSC-NCP complex, as the flexible bromodomains would be interacting with flexible histone tails, making them very flexible relative to the rest of the RSC-NCP complex.

We have cited Chatterjee et al. (see citation number 24).

We can make Figure 3 larger by moving panel c below a and b, and we will adjust the font size to improve legibility.

We have remade/rearranged the figure entirely. The figure captions are hopefully large enough now.

Reviewer #2:

A structure of the RSC complex is presented. This reveals a tripartite core made up of the Sth1, RSC8, RSC9, Sfh1, Rsc6 subunits, which are conserved in evolution. As a result, the findings are likely to inform structure of human complexes that play important roles in a range of human diseases.The structure refers to lower resolution negative stain structure of the SWI/SNF complex. From this it is concluded that SWI/SNF is likely to lack the tail lobe. As the tail lobe is not well resolved in RSC, an alternative is that it is dynamic in SWI/SNF and not resolved.

This is true. However, we can include 2D-class averages where we can see a tail density for RSC and not SWI/SNF. The reason we do not see the tail density in 3D is because we did not sort for it, and as it is sub-stoichiometric, it is only observable at much lower density thresholds. We can however do the sorting in 3D to show this.

The structure of the complex is used to infer the pathway by which it is assembled. It is incorrect to do this. No measurements have been made of the timing with which nascent proteins associate, and assembly intermediates predicted to arise in the absence of different subunits have not been identified. The structure defines interfaces that indicate how the complex is held together, not how it is assembled, and the manuscript should be re-written to reflect this.

We can change “The pattern of protein‐protein interactions observed in our structure provides insight into RSC assembly” to “Based on the observable protein-protein contacts in our structure of RSC, along with previous mass spectrometry analysis of the assembly of human BAF complexes, we propose a model for RSC assembly.”

We chose to build off the model that was proposed in a recent paper from the Kadoch lab (citation 16: Mashtalir, N. et al.). In their study, Mashtalir et al. did not investigate how nascent proteins associate, but they did identify “assembly intermediates predicted to arise in the absence of different subunits” by knocking out a subunit and analyzing the resulting complexes by mass spectrometry. What our work adds to theirs is subunit detail as we can see which subunits are interacting and suggest what their model would look like in atomic detail. This needs to be made clearer in the revised manuscript.

A low-resolution structure of RSC bound to a nucleosome is also provided. This should be submitted as a separate EMDB submission with statistics. Given the low resolution, it is highly speculative to describe interaction surfaces for components of the complex, such as bromodomains and the protein surfaces that contact nucleosomes. The nucleosome complex is used to propose a mechanism for remodelling involving, for example, interactions with histone acetylation marks. However, the manuscript adds nothing new in this respect as none of the interactions with acetylated histones are resolved.

We will be depositing the RSC-NCP map to the EMDB and can include an FSC curve to Figure 1—figure supplement 8. We do not state that we observed bromodomain density nor do we fit any bromodomains. What we do see is the ARP module and NCP, which we can dock unambiguously.

We are proposing a model of nucleosome engagement, not remodeling. What we are proposing is that RSC bind the nucleosome first through the acetylated tails and then through contacts with the NCP (such as the acidic patch).

We have added two key words to indicate the speculative nature of our model (in italics).

RSC *could* then engage the nucleosome…

The ATPase domain *would* then be able to translocate the DNA around…

The penultimate four paragraphs (prior to the final summary paragraph) are highly speculative. The two paragraphs before these incorrectly infer a pathway for the assembly of the complex.

As we have stated above, we can modify the opening statement of the paragraph discussing assembly to clarify where the model of assembly comes from (citation 16: Mashtalir, N. et al.).

The manuscript includes important new data describing how the core subunits of the RSC complex are organised, but a manuscript has been prepared with little attention to this strength.

We have significantly changed/expanded the content of our manuscript to include more discussion on the analysis of the structure. We now have dedicated sections for describing the head body and arm lobes of RSC and the roles each plays in RSC function.

Reviewer #3:

Chromatin remodelers are essential protein complexes that modify the position of nucleosomes in chromatin. The SWI/SNF family of remodelers have been studied extensively over the past several decades and are of high biological importance. While low resolution reconstructions of SWI/SNF modelers have been determined, the only high-resolution data that exists for SWI/SNF remodelers are of the ATPase domain alone and bound to the nucleosome.Here, Patel et al. used cryo-electron microscopy to determine the structure of apo RSC to near atomic resolution, as well as a low-resolution structure of RSC bound to the nucleosome. The overall structure can be subdivided into 5 distinct modules that include portions of 14 of its 17 subunits. Subunit assignments were verified by mass spectrometry crosslinking experiments. The main body of the structure is constructed around a Rsc8 dimer, and the authors put forth a model for how RSC assembles into a complex. Surprisingly, the ATPase domain is not resolved in either structure, suggesting that the ATPase-containing module is extremely flexible. The structure of RSC is a seminal achievement and will not only shed new insights on decades of past biochemical research but be informative for future studies on SWI/SNF remodeling.Overall, there are no major issues that need to be addressed. I would recommend several minor changes to the manuscript.TextSWRIM should be SWIRM.

We thank the reviewer for pointing out this typographic error, which will be corrected.

The authors introduce RSC as an important complex with broad functional applications, but do not mention that its specific function is to translocate DNA around the nucleosome until later in the text. For the sake of the general audience, it would be helpful if the authors briefly described what RSC does and what sets it apart from the other chromatin remodeling families.

A brief description of the function of RSC will be added.

There is no Figure 1E.

The reference to figures at this point should just state Figure 1D. We thank the reviewer for spotting this error.

Subsection “Structure of RSC and RSC-NCP”: “Our negative stain analysis of the yeast SWI/SNF complex shows that, like RSC, it has…. RSC features additional regulatory domains (Figure 1—figure supplement 2).” This point should be moved to where Rsc3 and Rsc30 are discussed so that it can be developed further there.

This can be done.

Figure 3D doesn’t exist but is referred to multiple times throughout the text.

This will be corrected, to Figure 3C.

Snf5 should be referred to as Sfh1 in the RSC complex.

We are referring to a domain of Sfh1 called Snf5. We see this can be confusing and can change it to clarify the nomenclature.

“Figure RSC-NCP2” Properly label.

This should be removed.

Epicypher nucleosomes often only have 147 bp of DNA wrapped around them. Do your nucleosomes have sufficient extranucleosomal DNA that this interaction would be possible?

While other studies have used nucleosome with extranucleosomal DNA, the ATPase domain binds to SHL+2 which would be within the 147 bp-region. However, we cannot fully exclude that the reason we do not observe the ATPase is because we are lacking this extra DNA.

FiguresFigure 3C: This figure needs to be reworked so that the RSC:NCP interactions are highlighted clearly. In the legend, the four contacts points should appear in the same order as they appear in the text. It would also be useful if they were numbered (i, ii, iii, etc) on the structure to easily locate the area being referred to.

These changed can be made.

Figure 1—figure supplement 3: Purple and grey volumes should be differentiated in the figure legend text.

A sentence stating that the purple classes indicate the classed that were selected for further processing will be added.

Figure 1—figure supplement 7: Figure needs a title.

The title “RSC core model validation” will be added.

Figure S5: 2D class averages should be included.

This will be added. Along with a micrograph.